# Modeling Reactive Oxygen Species-Induced Axonal Loss in Leber Hereditary Optic Neuropathy

**DOI:** 10.3390/biom12101411

**Published:** 2022-10-02

**Authors:** Darius W. Lambiri, Leonard A. Levin

**Affiliations:** 1Department of Physiology, McGill University, Montreal, QC H3G 1Y6, Canada; 2Department of Ophthalmology and Visual Sciences, McGill University, Montreal, QC H4A 3S5, Canada; 3Department of Neurology and Neurosurgery, McGill University, Montreal, QC H3A 2B4, Canada

**Keywords:** Leber hereditary optic neuropathy, mitochondrial disease, axonal degeneration, visual fields, optic nerve simulation

## Abstract

Leber hereditary optic neuropathy (LHON) is a rare syndrome that results in vision loss. A necessary but not sufficient condition for its onset is the existence of known mitochondrial DNA mutations that affect complex I biomolecular structure. Cybrids with LHON mutations generate higher rates of reactive oxygen species (ROS). This study models how ROS, particularly H_2_O_2_, could signal and execute the axonal degeneration process that underlies LHON. We modeled and explored several hypotheses regarding the influence of H_2_O_2_ on the dynamics of propagation of axonal degeneration in LHON. Zonal oxidative stress, corresponding to H_2_O_2_ gradients, correlated with the morphology of injury exhibited in the LHON pathology. If the axonal membrane is highly permeable to H_2_O_2_ and oxidative stress induces larger production of H_2_O_2_, small injuries could trigger cascading failures of neighboring axons. The cellular interdependence created by H_2_O_2_ diffusion, and the gradients created by tissue variations in H_2_O_2_ production and scavenging, result in injury patterns and surviving axonal loss distributions similar to LHON tissue samples. Specifically, axonal degeneration starts in the temporal optic nerve, where larger groups of small diameter fibers are located and propagates from that region. These findings correlate well with clinical observations of central loss of visual field, visual acuity, and color vision in LHON, and may serve as an in silico platform for modeling the mechanism of action for new therapeutics.

## 1. Introduction

In 1988, in a first for genetically linked disorders, Wallace et al. [1] described the potential link between a rare optic neuropathy and a mutation in mitochondrial DNA (mtDNA). The disease, known as Leber hereditary optic neuropathy (LHON), eponymously named after the physician who first described it in 1873 [2], is a neurodegenerative disorder that causes bilateral blindness by causing the death of retinal ganglion cells (RGCs). Although it can occur at any age, it primarily affects young adults [3]. LHON is a clinical diagnosis that begins with the onset of a cecocentral scotoma, though it is now definitively assessed by DNA testing for known mtDNA mutations [4]. Once started, vision loss happens rapidly over weeks or months, and although it usually begins in one eye, the contralateral eye inevitably follows [4]. All LHON cases have predominately homoplastic mitochondrial DNA point mutations that affect complex I subunits [1,5,6]. Three mutations are most prevalent (m.11778G > A in ND4, m.14484T > G in ND6, and m.3460G > A in ND1) and account for 95% of cases [7]. No known nuclear DNA mutations cause LHON [8], yet interplay with nuclear genetic factors is likely [9]. The deleterious effects of mtDNA mutations have also been established in animal models through transferring mutant human mtDNA ND4 [10] or mutant mtDNA ND6 genes [11] to mice, which caused visual loss similar to the effects experienced by LHON patients. Although the genetic defects are a necessary precondition for the disease, as the mitochondrial mutations are present in all LHON-affected patients, not all carriers of the defects exhibit the illness [12]. Penetrance of the disease is low in female carriers of the mutations [13], and a mitochondrial biogenesis compensatory mechanism is present in both carriers and affected cases [12]. People that have any of the LHON mitochondrial mutations exhibit more copies of their mtDNA than control cases [12]. Affected individuals, although they show more mtDNA per cell than controls, exhibit less mtDNA than carriers, implying a reduced mitochondrial compensation capacity [12].

Much progress has been made in unravelling LHON pathophysiology, molecular and environmental triggers, but the causal chain leading to the illness and the roots of its sudden onset and progression are still unelucidated [4]. While neuroretinal effects of the disease are well established [14,15], and studies of postmortem optic nerve samples have been published [16], the actual dynamics of injury propagation in RGCs and the biochemistry of the process that leads to RGC apoptosis are still unknown [7,17].

LHON-linked mtDNA mutations affect critical components of the mitochondrial electron transport chain, and, as a result, energetic imbalance and oxidative stress have been used to explain the death of affected RGCs [18,19]. The clinical findings in LHON (loss of visual acuity, cecocentral visual field loss, and dyschromatopsia) reflect the loss of small diameter axons, which led the Sadun group [16,17,20] to hypothesize that the axonal diameter predisposes the axon to death because of energy imbalance. In this hypothesis, and consistent with the axonal stress index (NFL-SI) that the Sadun group developed [16], axons undergo degeneration because of an imbalance between energy requirements, which are proportional to the axonal surface, and ATP production, which is proportionate to axonal volume. The axonal stress index (NFL-SI) is a good predictor of the bias towards injury of smaller diameter axons in LHON. However, the disease also has axonal loss topological characteristics that cannot be explained by the index alone. In particular, the ATP deficit hypothesis and the NFL-SI index can account for increased death susceptibility of smaller axons but does not account for the clustering of RGC loss seen in LHON optic nerve histology [21].

Experimental data from rats demonstrates that, in optic nerve axons, mitochondrial volume is directly proportional to the axon section volume in both myelinated and unmyelinated segments [22]. However, in cybrids with LHON mutations, overall cellular ATP production is normal, although ATP production per mitochondrion is impaired [23,24]. “However, in cybrids with LHON mutations, overall cellular ATP production is normal, although ATP production per mitochondrion is impaired [23,24]. Animal models with the 14484T > G (ND6) mutation, created by the Sadun group [24], and models with the 3460G > A (ND1) mutation, created by Zhang et al. [23], maintain levels of ATP production but cellular ROS levels increase. Maintaining overall cell ATP production can be explained by a mitochondrial compensatory mechanism [12], where the mitochondrial volume increases, compensating for the lower output per mitochondrion.

The role of reactive oxygen species (ROS) in the genesis and evolution of LHON is a promising alternative to the ATP deficit hypothesis [8,25]. Mitochondria are the site of oxidative phosphorylation and the source of most cellular superoxide ($O_{2}^{-})$ production [26,27], a precursor to all ROS compounds [28]. Although ROS compounds have intracellular signaling roles in fundamental cellular processes [29,30], it has also been recognized that higher concentrations lead to oxidative stress and can trigger cell death [31]. We used both $O_{2}^{-}$ and hydrogen peroxide (H_2_O_2_) as a stress-inducing species, recognizing that H_2_O_2_ physical properties make it a more likely candidate as the oxidative stress species. The range of ROS concentrations with beneficial physiological outcomes is relatively narrow. For example, H_2_O_2_ concentrations ([H_2_O_2_]) between 1 nM to 10 nM benefit neuronal development, while concentrations below 1 nM and over 100 nM led to tissue pathology [32].

There is experimental evidence that increases in ROS concentration are involved in LHON pathophysiology, although the actual mechanism mediating this process is unknown. For example, in animal models, ROS production increases by 48% when the 14484T > G (ND6) mutation is present [24], and by about 30% when the 3460G > A (ND1) mutation is present [23]. Data also points to lower antioxidant defenses in LHON cases, as exhibited by cybrids carrying LHON mutations, which can also lead to increased ROS [19]. It has also been shown that cigarette toxicity, often considered an environmental factor in the emergence of LHON, reduces ROS scavenging capacity [13]. Searches for therapeutic measures to prevent or stop its development have also concentrated on antioxidants and oxidative stress-reducing medication, although other approaches targeting estrogen receptors [33] and gene therapy have recently been studied [34]. Currently, the only marketed LHON therapy (authorized under exceptional circumstances in Europe) is idebenone, a synthetic analogue of CoQ10 that reduces oxidative stress [34], although the mechanism by which this action is achieved is unclear [35,36].

The basis for the present study is the hypothesis that LHON axonal degeneration has specific characteristics that might be explained by ROS spreading from one axon to another. Analysis of axonal diameter loss shows that, although smaller diameters are affected disproportionately, the effect highly depends on the optic nerve areas where the loss occurs [16]. Small optic nerve fibres are alive in unaffected areas, while larger diameter RGCs are dead in affected areas, which show complete RGC loss [16]. In other words, the loss of axons is area-specific and not solely diameter-specific.

We previously modeled the wave-like propagation of LHON [37], with a good correlation between clinically observed patterns of RGC loss and the size-dependent model presented by the Sadun group [16]. Our previous model [37] operated on a 2-dimensional (2D) section of the optic nerve and studied the propagation of axonal degeneration in the optic nerve from a selected initial point of injury. The present study extends the modeling from [37] 2D to 3 dimensions (3D), adds axonal membrane properties, a quantitative biochemical model, a more complex biological model for axonal sections, and the ability to generate ROS outside the axonal space. The 3D model also allows the study of processes around the nodes of Ranvier, present in the myelinated part of the optic nerve [38], calculates the time spent in the sample’s biology, allowing us to judge the timescales of the processes under study. With these enhancements, we studied the connection between the mitochondrial volume increases that are present in carriers and afflicted LHON patients [12]. We also studied under what conditions axonal fibers degenerate independently and when degeneration is zonal. Finally, we investigated possible triggers for the wave-like propagation of the illness. Both $O_{2}^{-}$ and H_2_O_2_ were used as oxidative agents. Finally, an unresolved question is the location of the initial injury for LHON along the optic nerve. The present study explored if the injury starts in the unmyelinated prelaminar region or the myelinated region of the optic nerve, using a computational approach.

## 2. Methods

### 2.1. Biochemical Modeling of Optic Nerve Reactive Oxygen Species

#### 2.1.1. Background and Rationale

Reactive oxygen species (ROS) have long been the focus as a possible cause of various physiological conditions and illnesses [27,39]. Mitochondria generate them during oxidative respiration and other intracellular and extracellular sources [26,29]. The chemical transformations from the primordial ROS species, $O_{2}^{-}$, to several derived chemical compounds, are established [28]. The loci for ROS production and their effects on organelle and tissue pathophysiology, particularly for neurodegenerative disorders, are areas of active research [40,41].

Multiple ROS species are derived from $O_{2}^{-}$ of which H_2_O_2_ is the most prevalent [42]. Our proposed model simplifies the full biochemical ROS transformation models [28,29,42] and only considers $O_{2}^{-}$ and H_2_O_2_ as ROS species of interest (Figure 1), with dismutation of $O_{2}^{-}$ to H_2_O_2_ by mitochondrial superoxide dismutase 2 (SOD2) [43].

Excess concentrations of H_2_O_2_ induce pores that permit the efflux of cytochrome *c* [44]. The model used local H_2_O_2_ concentrations thresholds ([H_2_O_2_]) in axonal segments as the element that forces mitochondria to release mediator(s), which induce axonal degeneration (Figure 1). We hypothesized that local H_2_O_2_ concentrations vary significantly in the axon and that oxidative stress in relatively small areas can trigger this mitochondria-based axonal degeneration.

#### 2.1.2. Reactive Oxygen Species Diffusion

The two ROS species’ chemical dynamics were modeled using first-order differential equations. The starting point of the derivations was a continuous-time representation of single species reaction-diffusion systems of the form described by Equation (1) [45], where [C] represents the concentration of a species, f([C], R) is the reaction function (R represents other reactants), and D is a diffusion tensor.
(1)$$\partial_{t}\left[C\right]=\nabla \cdot \left(D\nabla \left[C\right]\right)+f\left(\left[C\right],R\right)$$

Equation (1) was discretized in space and time to obtain a computable form, accounting for the permeability of the axonal membrane and optic nerve cross section anisotropy. All points in the model are initialized to the same concentration value c_0_.

ROS concentrations ([ROS]) were tracked at each (i, j, k) voxel position. There were 3 different operating equations for each species: Equations (2)–(4) modeled $O_{2}^{-}$ dynamics, while Equations (5)–(7) modeled H_2_O_2_ dynamics. Only one equation is used at a voxel.

The simulation environment implemented only one set of 3 equations. Given that Equations (2)–(7) are similar, it was possible to simulate both $O_{2}^{-}$ dynamics or H_2_O_2_ dynamics in the tissue using a single system implementation. However, we focused on H_2_O_2_ dynamics because this species is less labile and has a longer range signaling capacity [46].

Voxels that modeled mitochondria used Equation (2) or Equation (5). All non-mitochondrial intracellular voxels used Equation (3) or Equation (6). Intercellular spaces not occupied by intercellular mitochondria used Equation (4) or Equation (7).
(2)$$\;\;\;\;\;\frac{d{\left[O_{2}^{-}\right]}_{i,j,k}}{dt}=\nabla \cdot \left(D_{O_{2}^{-}i,j,k}\nabla^{2}{\left[O_{2}^{-}\right]}_{i,j,k}\right)-k_{s2}{\left[SOD_{2}\right]}_{i,j,k}{\left[O_{2}^{-}\right]}_{i,j,k}+k_{p}{\left[CI\right]}_{i,j,k}{\left[O_{2}\right]}_{i,j,k}$$
(3)$$\;\;\;\;\;\frac{d{\left[O_{2}^{-}\right]}_{i,j,k}}{dt}=\nabla \cdot \left(D_{O_{2}^{-}i,j,k}\nabla^{2}{\left[O_{2}^{-}\right]}_{i,j,k}\right)-k_{s1}{\left[SOD_{1}\right]}_{i,j,k}{\left[O_{2}^{-}\right]}_{i,j,k}$$
(4)$$\;\;\;\;\;\frac{d{\left[O_{2}^{-}\right]}_{i,j,k}}{dt}=\nabla \cdot \left(D_{O_{2}^{-}i,j,k}\nabla^{2}{\left[O_{2}^{-}\right]}_{i,j,k}\right)-k_{s3}{\left[SOD_{3}\right]}_{i,j,k}{\left[O_{2}^{-}\right]}_{i,j,k}+Prod_{inter,i,j,k}$$
(5)$$\frac{d{\left[H_{2}O_{2}\right]}_{i,j,k}}{dt}=\nabla \cdot \left(D_{H_{2}O_{2}i,j,k}\nabla^{2}{\left[H_{2}O_{2}\right]}_{i,j,k}\right)-k_{sh2}{\left[GPX\right]}_{i,j,k}{\left[H_{2}O_{2}\right]}_{i,j,k}+k_{s2}{\left[SOD_{2}\right]}_{i,j,k}{\left[O_{2}^{-}\right]}_{i,j,k}$$
(6)$$\frac{d{\left[H_{2}O_{2}\right]}_{i,j,k}}{dt}=\nabla \cdot \left(D_{H_{2}O_{2}i,j,k}\nabla^{2}{\left[H_{2}O_{2}\right]}_{i,j,k}\right)-k_{sh1}{\left[GPX\right]}_{i,j,k}{\left[H_{2}O_{2}\right]}_{i,j,k}+k_{s1}{\left[SOD_{1}\right]}_{i,j,k}{\left[O_{2}^{-}\right]}_{i,j,k}$$
(7)$$\frac{d{\left[H_{2}O_{2}\right]}_{i,j,k}}{dt}=\nabla \cdot \left(D_{H_{2}O_{2}i,j,k}\nabla^{2}{\left[H_{2}O_{2}\right]}_{i,j,k}\right)-k_{sh3}{\left[CAT\right]}_{i,j,k}{\left[H_{2}O_{2}\right]}_{i,j,k}+k_{s3}{\left[SOD_{3}\right]}_{i,j,k}{\left[O_{2}^{-}\right]}_{i,j,k}$$

The above equations were implemented in the simulation from user-specified inputs. ROS production (RP), e.g., $RP=k_{s2}{\left[SOD_{2}\right]}_{i,j,k}{\left[O_{2}^{-}\right]}_{i,j,k}$ in Equation (2), was user-specified in µM/s. ROS scavenging (RS), such as $RS_{Sh2}=k_{sh2}{\left[GPX\right]}_{i,j,k}$ in Equation (5), was user-specified in 1/s.

#### 2.1.3. Oxidative Stress States

Physiological H_2_O_2_ concentrations were divided into 2 overlapping ranges, *oxidative distress* and *oxidative eustress* [47]. Neurons, like other cells, enter oxidative distress at H_2_O_2_ concentrations over 100 nM and enter oxidative death at H_2_O_2_ concentrations over 1 µM [32]. However, H_2_O_2_ concentrations have gradients in the simulation model that limit the use of an average model concentration. Concentrations averaged over the entire model are therefore denoted as [H_2_O_2_]_MODEL_.

Axonal segments were assigned one of 3 states to account for the concept of oxidative distress: “Healthy” [H], “Oxidative Stress” [S] (which corresponds to oxidative distress), or “Dead” [D] (Figure 2) each of which could have different ROS production, scavenging and membrane diffusion coefficients. ROS production ceased in the [D] state. The transition into the [D] state abstracts the moment when mitochondria stop functioning permanently. Transitioning to the [D] state for axonal segments marks the loss of mitochondrial function. State changes between the [H], [S], and [D] states (Figure 2) were based on concentration thresholds or time spent in the [S] state. Specifically, the [D] state could be reached in two ways: if the ROS concentrations ([ROS]) reach a user-specified threshold, for example, 1 µM [32]; or if the axonal segment was exposed for sufficient time to sufficient [ROS] to cause oxidative stress but below the concentration known to induce irreversible axonal degeneration outright. Our model allowed testing of the hypothesis that axonal degeneration starts when the axon is under oxidative stress conditions for extended periods.

Mitochondrial in glia were placed in the inter-axonal space but without demarcating glial cell boundaries. Each glial mitochondrion could be in one of 3 states (Figure 2), similar to axonal mitochondria. For glial mitochondria, the transition to the [D] state marked the mitochondrial death and cessation of its ROS production.

All voxels in an axonal segment were assigned the same state. This assumption limited the 3D axonal mitochondria topologies that could be simulated because mitochondria are dispersed along the axon, and [ROS] can vary substantially along the axon. At the nodes of Ranvier, mitochondria are present next to but not at the node [22]. Energetic requirements are dominated by signal conduction. Nearby mitochondria support the nodes to reduce the time for diffusion of ATP, while mitochondria that are farther from the node in the internodal region will have lower energy requirements [48]. The model implements this energetic requirement by allowing different ROS production and scavenging values along the axon. Setting thresholds only for the segments closer to the node is sufficient because those segments will experience higher oxidative stress due to higher requirements for ATP.

### 2.2. Topological Modeling of Optic Nerve Axons and Mitochondria

#### 2.2.1. Modeling Axons

The complex geometry of the optic nerve (ON) makes three-dimensional (3D) modeling of the entire ON difficult. The current work simplified the problem by restricting modeling to axonal sections in the region, starting at the optic nerve head (ONH) and extending to but not including the chiasm. In the model, all axons are parallel, a reasonable abstraction for small optic nerve sections.

Samples were mapped to 3D meshes, with equal mesh steps in the X and Y dimensions, while the Z dimension mesh step could be set independently. Thus, each point in the mesh is a voxel corresponding to a rectangular cuboid of the optic nerve. For example, if the XY resolution was 7 pixels/µm and Z resolution was 1 pixel/µm, each pixel represented a physical cuboid of sizes d_x_ = d_y_ = 1/7 µm, d_z_ = 1 µm, and volume V = 1/49 µm^3^.

In the optic nerve, RGC axon diameter distributions differ between optic nerve areas [16]. To model axonal placements in the ON, the stochastic placement algorithm created by Coussa et al. [37] was used, reproducing the ON axonal distributions reported by Pan et al. [16]. The algorithm models axons as non-overlapping circles and places them in a representation of an ON. In out implementation, the algorithm was extended to account for the axonal membrane by accounting for the voxels that formed the perimeter of the axons and given these voxels membrane-specific properties.

Healthy young adult optic nerves contain 1.2 million RGCs [49]. Models that include such numerous axons are computationally demanding. If *diam_ON_* denotes the optic nerve diameter, then an *“x”%* model of that optic nerve is a model of diameter $diam_{ON}\frac{x}{100}$. Such reduced models have the same axonal diameter distributions as the full optic nerve. However, the number of axons in the model is reduced by approximately ${\left(\frac{x}{100}\right)}^{2}$ relative to a full optic nerve model.

#### 2.2.2. Modeling Axonal Mitochondria

Axonal mitochondria have a tubular-like appearance with diameters between 50 nm and 300 nm [22,50] and lengths less than 2 µm [51]. The size in our model was set to 1 µm, close to the reported median [51]. Mitochondria were simulated as a “single volume” model, and all their biochemical properties were compacted in a single simulation voxel. By consolidating the mitochondrion to a single voxel, larger tissue-level models could be simulated at the expense of abstracting the intricacies of the mitochondria biochemistry. Compacting a mitochondrion in a voxel creates a modelling limitation for the geometries which can be supported, and necessarily links the dimensions of the mitochondrial model to the model resolution. This was done to improve throughput in the simulation procedure.

A stochastic algorithm determined the mitochondrial placement within axons by creating them with a probability *p(x < mito_axon_)*, where *mito_axon_* was a user-defined percentage of the axonal mitochondrial volume ratio. The same algorithm was employed to place glial (extra-axonal) mitochondria. Mitochondria were placed in the optic nerve volume not occupied by axonal mass with a probability *p(x < mito_glia_),* where *mito_glia_* represents the desired mitochondria percentage of the overall extra-axonal volume.

Unmyelinated and myelinated axonal regions

The study modeled both myelinated and unmyelinated regions of the ON. Unmyelinated RGC axonal segments contain greater mitochondria concentrations, reflecting their increased metabolic rate imposed by axonal conduction requirements [22,52]. In the unmyelinated region, given that an axon’s energetic needs do not change along the course of the axon (i.e., the Z-axis), the use of models with one or multiple stacked voxels in the Z dimension did not change the simulation results. Thus, all simulations for unmyelinated regions were performed with a single voxel in the Z dimension to reduce computational load. Myelinated axons were modeled as if they contained a single node, midway between the proximal and distal extents of the axon. The node’s length in the Z dimension was set to 1 µm, the median of reported values [53].

### 2.3. Visualization of Simulation Processing and Results

The simulation software contained a visualization module which displayed ROS concentrations and axonal states while running simulations. 3D ROS concentrations could be examined through 2D cuts in the 3 anatomical planes: coronal, transverse and sagittal (Figure 3). Coronal plane views are split into 8 zones called “octants”: *temporal* (T) octant on the left, the *nasal* (N) octant on the right, the *superior* (S) octant at the top, and the *inferior* (I) octant at the bottom. In other words, a right optic nerve viewed from the front was always modeled.

Axons were also grouped into 8 octants for data processing, each covering a 45° angle circle sector. From left to right in the clockwise direction, the octants were *temporal* (T), *superior-temporal* (ST), *superior* (S), *superior-nasal* (SN), *nasal* (N), *inferior-nasal* (IN), inferior (I) and *inferior-temporal* (IT) (Figure 3). In each of the eight coronal octants, axonal loss is measured as *Loss_Octant_ = max*(10 × log_10_(*N_Alive[Octant]_/N_Total[Octant]_*), −101) and measured in dB. The lower limit for loss was selected because the logarithm would be minus infinity when the loss in an octant is complete. A lower limit was needed for display, and the number (−101) chosen is less than the dB value if only one axon in an octant survives.

### 2.4. Model Calibration

#### 2.4.1. Diffusion and Permeability

Each voxel had a three-dimensional diffusion vector (D_xx_, D_yy_, D_zz_). Diffusion values for the X and Y directions were always equal (D_xx_ = D_yy_), while the diffusion in the Z direction could be set independently. Thus, the model space was anisotropic.

Membrane permeability is related to diffusion through the formula P_m_ = D_m_/L [54], where P_m_ is the permeability of the species, and L denotes the thickness of the membrane. Cellular membrane thickness varies between 4 to 10 nm [55,56], while the myelin thickness is about 25% of the axon diameter in data from the optic nerve of mice [57]. In our models, given that membranes are simply voxels, and the only medium property is diffusion, permeability was modeled by setting diffusion in the membrane voxels independently from the rest of the model. All diffusion voxels were constrained to D_zz_ = 0 µm^2^s^−1^, i.e., diffusion through the membrane voxels was only in the XY plane. In the rest of this paper, membrane diffusion values, denoted by D_m_, represent diffusion constants in the X and Y directions for membrane voxels.

The diffusion coefficient in water for $O_{2\;}^{-}$ is D_m_ = 2000 µm^2^s^−1^ [54]. Because of their similar molecular mass values, the same diffusion coefficient value was used for H_2_O_2_ and $O_{2}^{-}$ in simulations.

Thus, the diffusion coefficients in the membrane voxels were set to:(8)$$\;\;\;\;\;\;\;\;\;\;\;\;\;\;\;\;\;\;\;\;\;\;\;\;\;\;\;\;\;\;\;\;\;D_{m\;}=P_{m}*L=\frac{P_{m}}{res_{xx}}$$

Reported values for passive permeability of H_2_O_2_ through cellular membranes vary. Literature searches have yielded no values for neurons. Cellular membrane permeability to H_2_O_2_ is between 2.8 µm s^−1^ and 16 µm s^−1^ [58,59]. However, active transport is the primary mechanism for water permeability in astrocytes, with a reported water permeability of 500 µm s^−1^ [60]. Thus, reported H_2_O_2_ permeability constants may underestimate actual species transport capabilities for axons because of the large density of ion channels in the axonal membrane. As the reported values are in a large range, with values for axons unknown, the current work explores H_2_O_2_ diffusion values in the range from 2/res_xx_ µm^2^s^−1^ to 200/res_xx_ µm^2^s^−1^. The permeability decreases in myelinated segments because the myelin comprises many cellular membranes wrapped around the axon. An average myelin thickness of 0.2 µm [57] and a membrane thickness of 10 nm results in an average of 10 myelin folds around the axon. Thus, through myelin, there is a 20× reduction in H_2_O_2_ permeability.

Membrane permeability to $O_{2}^{-}$, measured to be around J = 2.1 × 10^−2^ µm s^−1^ [61] is so small that most references quote it as zero [62]. The resolution of 10 pixels/µm corresponds to a diffusion value of D_m_ = 2.1 × 10^−3^ µm^2^s^−1^. As there is no difference in simulation results when using D_m_ = 2.1 × 10^−3^ µm^2^s^−1^ or the results D_m_ = 0 µm^2^s^−1^ results sections will use D_m_ = 0 µm^2^s^−1^ as the membrane diffusion for $O_{2}^{-}$.

#### 2.4.2. Reactive Oxygen Species Production

The current work explores two hypotheses regarding mitochondrial ROS production. In the baseline hypothesis, ROS production per unit of mitochondrial volume is independent of the axon characteristics. Hence, if the mitochondria have the same volume, they will produce equal amounts of ROS. Simulations that use this hypothesis are marked with RP_SAME_.

In the constant neuronal firing frequency hypothesis, the amount of ROS produced in a unit of time by mitochondria is related to ATP production because mitochondrial ROS generation is relative to electron flow down the mitochondrial respiratory chain. Because the total amount of ATP produced is proportional to the energetic requirements of the segment, if more mitochondria are present, each mitochondrion will need to produce fewer ATP molecules per unit of time.

The energy needs of an axon are dominated by signal conduction [22], and the conduction energy requirements per unit of axonal length (L) are proportional to the axon surface area (2πRL), the total ROS produced per unit of time and unit of axonal length will be proportionate to the axonal membrane area and the neuron firing frequency (f). In the unmyelinated region, the volume of mitochondria in a unit of axonal length is πR^2^LxMito_axon%_. Thus, per unit of volume, the mitochondria will produce ROS proportional to:(9)$$\;\;\;\;\;\;\;\;\;\;\;\;\;\;\;\;\;\;\;\;\;\;\;\;\;\;\;\;\;\;\;\;ROS\propto \;\frac{f_{neuron\_fire}}{R\times Mito_{axon\%}}$$

Therefore, the production rate for mitochondria in an axon of radius R is inversely proportional to the axon radii:(10)$$\;\;\;\;\;\;\;\;\;\;\;\;\;\;\;\;\;\;\;\;\;\;\;\;\;\;\;\;\;RP_{H,S}\left(R\right)=\frac{\alpha \times f_{neuron\_fire}}{R\times Mito_{axon\%}}$$

Under the constant neuronal firing frequency hypothesis for two axons of radii R_1_ and R_2_, which fire at the same frequency f, and for axonal segments where the Mito_axon%_ values are the same, mitochondrial ROS production in two axons is related through Equation (11).
(11)$$\;\;\;\;\;\;\;\;\;\;\;\;\;\;\;\;\;\;\;\;\;\;\;\;\;\;\;\;RP_{H,S}\left(R\right)=\frac{R_{2}}{R_{1}}RP_{H,S}\left(R_{2}\right)$$

Simulations conducted under the constant firing hypothesis were marked with RP_FCONST_.

Sadun et al. [16] proposed a novel LHON axonal degeneration predictor that they named the axonal stress index (NFL-SI). The index resembles Equation (9) in that there is inverse proportionality with the axonal radius. Applying NFL-SI framework to the mitochondria, we can observe that, in the RP_FCONST_ hypothesis, mitochondria in smaller axons will experience greater oxidative stress.

One goal of our study was to explore if ROS generation rates that change as a function of the overall ROS concentrations affect general loss patterns. It is unclear, from available data, if the mitochondria ROS generation rate remains the same when it is subject to oxidative stress conditions. We group the ROS Stress state production behaviors into two categories as defined below. By running simulations with both hypotheses and comparing the results against known patterns of axonal loss, we can determine if either of these behaviors is more probable in LHON inception and propagation. As, in extremis, raising the ROS production rates can lead to very high ROS concentrations, well above pathological bounds, we have limited both the decreases and the increases in ROS productions to 30% of the RP_H_ values.
If the overall ROS generation in the Stress state is decreased (RP_S_ = 0.7 × RP_H_), the simulation was annotated with NF_RP_.If the overall ROS generation in the Stress state stays the same or increases (RP_S_ = RP_H_ or RP_S_ = 1.3 × RP_H_), the simulation was annotated with PF_RP_.


### 2.5. Simulation Platform

The simulation program was coded in C# and C [63] and was developed using Microsoft Visual Studio 2019 [64] and CUDAv11 [65]. An open-source package, Cudafy.Net [66], was used to connect the C# environment and CUDA. Simulations were executed on a PC equipped with an NVIDIA RTX2060 GPU containing 6 GB of onboard memory.

## 3. Results

### 3.1. Superoxide Alone Cannot Explain LHON Injury Patterns

Superoxide is the primary ROS generated by complex I, and its production is increased in LHON cybrids [23,24]. However, $O_{2}^{-}$ is an electrically charged species that does not diffuse through cellular membranes [61] and can be dismutated to H_2_O_2_. To test if $O_{2}^{-}$ can explain the zonal loss patterns observed in LHON pathologies, we ran simulations that generated $O_{2}^{-}$ only intra-axonally and did not allow $O_{2}^{-}$ to migrate to the inter-axonal space (i.e., Pm set to 0 µm s^−1^). We set the $O_{2}^{-}$ concentration threshold at which axonal degeneration would occur to 10^−1^ nM because reported cellular concentrations of $O_{2}^{-}$ are in the 10^−2^ nM range [47], and there are no known concentration thresholds that would trigger pathological issues because $O_{2}^{-}$ is rapidly converted to H_2_O_2_ in vivo [42]. Given the lack of knowledge about whether stressed axons produced more or less ROS than health axons, we compared the results of simulations where there was increased (PF_RP_) or decreased (NF_RP_) $O_{2}^{-}$ production, to check the sensitivity of results to ROS production changes in the Stress state.

Loss patterns were similar irrespective of the mitochondrial ROS production patterns because axonal loss depends only on each axon’s ROS production and is not influenced by superoxide diffusion from other cells (Figure 4a,b). However, if all mitochondria produced ROS at the same rate (RP_SAME_), larger diameter axons were affected in higher numbers (Figure 4a simulations marked with RP_SAME_), which is uncharacteristic for LHON. The axonal loss was uniform across all optic nerve octants (Figure 4b), and at the end of simulations, each octant had more mid-diameter axons alive than small or large diameters. The loss patterns were also atypical for LHON [16,37]. Interestingly, although all simulations presented in Figure 4a had similar surviving axonal ratios (around −6dB for each octant), they had markedly different patterns in the coronal sections of the optic nerve, with the experiments where larger diameter axons degenerated showing a more sparse appearance (i.e., a lot more “blue” areas in Figure 4a) Although less numerous, larger diameter axons occupied significant portions of the overall optic nerve area. If they are affected, a significant portion of the optic nerve area loses neurons, and the remaining axons are sparsely distributed among areas where no other axons are present. When mitochondria produced $O_{2}^{-}$ at the same rates, larger diameter axons were disproportionately affected (Figure 4c,d).

When $O_{2}^{-}$ production uses the constant neuronal firing frequency hypothesis (RPF_CONST_), axonal loss is heavily correlated with the axonal diameter and skewed towards lower diameter axons. The affected axons demonstrated extreme diameter selectivity (Figure 4c), with smaller axons completely degenerating while larger diameters were unaffected. Thus, optic nerve octants were affected in proportion to their smaller diameter axons. Surviving and dead axons intersperse, with no area of complete axonal loss. The observed patterns differed from actual LHON axonal loss patterns, which show a total loss in large areas, regardless of axonal diameters [16,37].

Thus, given simulation results that could not replicate actual LHON injury patterns because of the inability of $O_{2}^{-}$ to diffuse across cell membranes, it is unlikely that $O_{2}^{-}$ is the primary axonal damage agent in LHON. Because of the permeability of H_2_O_2_ through cell membranes, subsequent simulations focused on it as a potential oxidative stress mediator for axonal degeneration in LHON.

### 3.2. The Constant Neuronal Firing Frequency Hypothesis Predicts Higher Oxidative Stress in the Temporal Optic Nerve

We considered two ROS production hypotheses, one in which all mitochondria produced ROS at the same rate (RP_SAME_), and one where mitochondria produced ROS in proportion to the axonal energy requirements (RPF_CONST_). To determine which of the two production models was more relevant to LHON, we ran simulations with identical initial topology and biochemistry but with different H_2_O_2_ production models and measured the oxidative stress produced by the mitochondria. Both unmyelinated (Figure 5a) and myelinated (Figure 5b) topological models were used for this purpose. Axonal mitochondria volume ratio (mito%) was measured as a proportion of the axonal volume it occupies [22]. Simulations were calibrated at 4% mito% in the unmyelinated region [22] so that, regardless of the ROS production strategy, mean H_2_O_2_ concentrations were the same (Figure 5a).

All simulations demonstrated the formation of nonuniform H_2_O_2_ concentration gradients in the nerve, ranging from 5–100 nM. Axonal mitochondrial volume ratios (mito%) influenced the gradients in ways that depended on the production model and the myelination of the region (Figure 5). When all mitochondria produced H_2_O_2_ at the same rate (RP_SAME_), H_2_O_2_ concentrations increased as mito% increased (Figure 5 and Figure 6). This phenomenon happened in both unmyelinated and myelinated regions because the total H_2_O_2_ produced was proportional to the total amount of mitochondria present when the rate of mitochondrial ROS production was fixed. Under this production model, carriers of LHON mutations would experience higher oxidative stress than LHON cases. Given that localized H_2_O_2_ concentrations can be higher than the mean concentration, many axons could experience high oxidative stress levels, increasing the probability of axonal degeneration (Figure 6c). The constant neuronal firing frequency assumption (RPFCONST) demonstrated inverse correlations between mean H_2_O_2_ concentrations and mito%.

Other differences between the RPFCONST and RPSAME production models were observed. Higher concentrations of $O_{2}^{-}$ in the temporal nerve relative to the nasal nerve were observed in all simulations that used the RPFCONST hypothesis (Figure 6), while in the RPSAME production hypothesis, these zonal gradients were undetectable for mito% above 4% (Figure 6c). The axonal diameter was positively correlated with oxidative stress in the RPSAME cases (Figure 6c) and inversely correlated in the RPFCONST cases (Figure 6c).

Increasing the axonal membrane permeability allowed higher rates of ROS diffusion between the axon and the extra-axonal space, which reduced [H_2_O_2_] gradients in unmyelinated and myelinated model simulations because it moved the H_2_O_2_ from its mitochondrial source in the axon to a larger scavenging volume. The differences between the mean ROS concentration and the distribution of individual voxel concentrations illustrate the challenge of using means as a figure of merit in this case. For example, H_2_O_2_ concentrations inside axons can be above the threshold for degeneration, yet the mean concentration across the entire simulated volume can be significantly below those values. Thus locally, H_2_O_2_ can damage axons, while the axons remain intact in other parts of the model.

Under the hypothesis that high H_2_O_2_ concentrations trigger axonal degeneration, the constant mitochondria production assumption (RP_SAME_) creates optic nerve ROS levels that would result in more extensive optic nerve damage when the mito% is increased. However, LHON carriers have higher mito% than LHON cases, even though both have the same mutation [12], implying a lower oxidative load on the optic nerve. Thus, the RP_SAME_ production model is unlikely because it does not fit the data from studying LHON axonal mitochondria volume ratios (mito%) and was not studied in subsequent simulations.

### 3.3. Unmyelinated Regions of the Optic Nerve Have Higher ROS Levels Than Myelinated Regions

The optic nerve has a narrow geometry, with a diameter less than 3% of its length. In geometries where one dimension is significantly larger than others, diffusion processes proceed faster on the smaller dimensions. Therefore, lengthy areas where scavenging is present without significant ROS production might reduce the influence that high oxidative stress regions can have on their surroundings. We hypothesized that H_2_O_2_ concentration values and their changes in time in the unmyelinated region are largely decoupled from the [H_2_O_2_] in the myelinated area.

To study the hypothesis, simulations were performed with an abridged optic nerve (X = 150 µm, Y = 150 µm, and Z = 410 µm) containing myelinated and unmyelinated regions. Along the Z axis, the top 50 µm modeled the unmyelinated region, followed by 200 µm axonal myelinated region, a 5 µm section comprising a node and another 155 µm region modeling a myelinated axonal shaft. The node’s length was larger than the published data from rats [53] because of limitations in the simulation setup for samples of this scale. The internodal sizes were between reported values [53]. Nodes from all axons were aligned in the same plane, which is not physiological, but serves to increase oxidative coupling between axons, thus creating a more oxidative environment than in vivo. Therefore, this limitation should not have changed the qualitative nature of the results.

In this simulation, the role of scavenging rates along the axon was also examined. For this purpose, either similar scavenging rates across the model (Figure 7) or a 30% of unmyelinated scavenging capacity in the myelinated regions (Figure 8) were assumed.

Simulations were run until [H_2_O_2_] reached a steady state. The unmyelinated regions had significantly higher [H_2_O_2_] than the myelinated regions for both uniform scavenging of H_2_O_2_ (Figure 7) and reduced H_2_O_2_ scavenging in the myelinated area (Figure 8). In both cases, coronal plane views (Figure 7a and Figure 8a) demonstrate [H_2_O_2_] gradients between the temporal and nasal sides, with higher [H_2_O_2_] on the temporal side of the nerve.

Transverse (Figure 7b and Figure 8b) and sagittal (Figure 7c and Figure 8c) views demonstrate significantly higher gradients along the Z axis than in the XY plane. H_2_O_2_ concentrations in the myelinated and the unmyelinated regions were higher than the internodal concentrations when scavenging enzymes were uniformly distributed in the sample (Figure 7b,c).

The mitochondria in the internodal region had lower H_2_O_2_ concentrations than the paranodal mitochondria, which support signal conduction by providing ATP to the node (Figure 7 and Figure 8). The decrease in internodal scavenging increased H_2_O_2_ concentrations in the internodal regions, while concentrations in other regions were similar and higher than in the internodal regions (Figure 8). Coronal views of the unmyelinated region show no change in H_2_O_2_ concentrations (top row of Figure 7a and Figure 8a), although a coronal section of the node shows a small increase (third row of Figure 7a and Figure 8a).

In both cases, H_2_O_2_ concentrations in the internodal regions are lower than the values reached in the unmyelinated region or at the node. As a result, the H_2_O_2_ gradients are towards the internodal regions, and no gradient connects the myelinated and unmyelinated regions. Therefore, decomposing the unmyelinated and myelinated regions into separate models would not affect results.

The results from these simulations enable the separation of a nerve into separate models of its myelinated and unmyelinated regions, running the simulations on the 2 models using the same biochemical setup and analyzing the results as if they were from a unified model. Such a separation is useful for simulation throughout because large models have high computational requirements.

### 3.4. Myelinated Regions Are Unlikely Locations for Initiating Axonal Degeneration in LHON

The region where LHON optic nerve lesions are initiated and the dynamics of axonal degeneration associated with the disease are unknown. Under the hypothesis that the disease is triggered by oxidative stress in the axonal region of the optic nerve, we asked if the myelinated or the unmyelinated regions are more prone to high oxidative stress. Intuitively, higher axonal mitochondrial volume ratios (mito%) in the unmyelinated region [22,52] should have higher concentrations of ROS. However, as shown previously, the ROS gradients within the confines of the axon can be very high and vary based on the axon diameter. We also showed that when mitochondria produce ROS in proportion with the axonal energy requirements, overall H_2_O_2_ levels decrease as the axonal mito% increases. The myelinated region membranes are also less permeable to H_2_O_2_ because of the many myelin layers surrounding them. From that perspective, the oxidative environment in the myelinated regions is greater. However, previous simulation results in the myelinated and unmyelinated models showed higher oxidative stress in the unmyelinated region. Thus, we hypothesized that mitochondria in the unmyelinated regions are more likely to suffer from higher levels of oxidative stress, potentially triggering axonal degeneration and subsequent neuronal apoptosis.

We tested this hypothesis on unmyelinated and myelinated models, running simulations that considered only the effects of H_2_O_2_ produced by axons while varying the mitochondrial H_2_O_2_ in the Stress state (Figure 9 and Figure 10). Myelinated region models were created as coronal slices around a node, with an overall model thickness of 10 µm. A single node was placed 6 µm from the proximal side of the model. Mitochondria were placed symmetrically above and below the node, with no mitochondria present at the node.

Signal conduction energy requirements represent approximately 75% of neuronal energy requirements, with the rest used for homeostatic cellular functions [67,68]. The myelinated segment of the optic nerve dominates the axon length, with internodal distances between 100–200 µm for each node of 1–2 µm [53]. Therefore, in axonal segments centered on a node, most internodal mitochondria produce 25% of the required ATP, while the mitochondria close to the node produce 75% of the energetic needs of the segment. Assigning an estimated 5% ROS production rate to the internodal mitochondria farther from the node should account for the energy requirements of that segment.

To further explore myelinated vs. unmyelinated axons, we again assessed two models for oxidatively stressed axons, one with increased oxidative production in the Stress state (PF_RP_) and one with decreased production (NF_RP_). Simulations were performed to test the possibility that mitochondria change their operating points when subject to oxidative stress. For the PF_RP_ cases, we ran simulations where the Stress and Healthy states had either the same production values (Figure 10a) or increased production values (Figure 10b). To test sensitivity to membrane permeability, we used two different H_2_O_2_ membrane permeability values (P_m_ = 20 µm s^−1^ and P_m_ = 200 µm s^−1^).

Regardless of H_2_O_2_ production changes in the Stress state, the proportion of axonal loss was significantly lower in simulations of myelinated axons than in unmyelinated axons (Figure 9 and Figure 10). Notably, there was no axonal loss in the myelinated regions in NF_RP_ simulations (Figure 9) or PF_RP_ simulations (Figure 10a). In PF_RP_ simulations with increased stress levels, myelinated regions experienced loss only for the lower membrane permeability value.

In unmyelinated regions, the higher energy needs imposed by less efficient axonal conduction led to higher H_2_O_2_ production rates. For unmyelinated models, even if H_2_O_2_ production was reduced in the Stress state (NF_RP_), axonal degeneration occurred across a wide range of mito% values (4% to 15%) (Figure 9). However, axonal survival increased linearly with mito% (Figure 9).

Simulations where production in the Stress state stayed the same or increased (PF_RP_) had greater variability in axonal survival than the NF_RP_ simulations for the same mito% (Figure 9 and Figure 10). For the PF_RP_ simulations, the proportion of surviving axons graphed against the mito% had a more sigmoid shape. The proportion of surviving axons changes more dramatically for small changes in mito% as the production of H_2_O_2_ increases in the Stress state (Figure 10a,b). At the higher membrane permeability value (P_m_ = 200 µm s^−1^) and for the case when production in the Stress state increases, there were abrupt changes in survival for small variations in mito% around the 10% mito% value (Figure 10b). In this case, slight topological changes resulting from re-running the axonal placement algorithm for each run result in greatly different axonal survival ratios. The sensitivity of results to small axonal placement changes suggests that the interplay between axonal topology, membrane permeability and mito% are relevant to oxidative injury to axons.

Membrane permeability increases oxidative linkage between adjacent axons. Surprisingly, higher membrane permeability increases axonal losses at lower mito% and decreases axonal loss at higher mito% (Figure 9 and Figure 10). This was true for both Stress state production models. Specifically, in the NF_RP_ production model, a higher membrane permeability decreases axonal loss when mito% > 5% (Figure 9), in the Stress = Health model when mito > 7% (Figure 10a), and in the PF_RP_ model when mito% > 10% (Figure 10b). The existence of a mito% value below which high membrane permeability is a deleterious factor in axonal survival and above which membrane permeability has a positive effect demonstrates that the many factors which contribute to the subcellular oxidative environment interact with each other in nonobvious ways. In this case, the interaction between membrane diffusion, optic nerve axonal topology, mito% and Stress state oxidative behavior nonlinearly affect simulation outcomes.

The simulation results suggest that myelinated portions of RGC axons are unlikely to be the starting point of the injury in LHON. It is more likely that the injury occurs in the unmyelinated portion of the optic nerve and that axonal membrane permeability to H_2_O_2_ creates an oxidative environment which leads to axonal loss. It is also probable that during the acute phase of the injury, mitochondria will produce the same or greater amounts of H_2_O_2_, thus intensifying the overall oxidative stress environment of the nerve.

### 3.5. Increasing Physiological Coupling between Axons Increases Localized Axonal Degeneration

Results from previous sections show that axonal degeneration was strongly correlated with axon diameter when the axonal membrane was impermeable to ROS (as with $O_{2}^{-}$). Each coronal optic nerve octant had surviving axons (Figure 4b). However, if a ROS had sufficient membrane permeability to diffuse through the axonal membrane (as with H_2_O_2_), different axonal degeneration patterns occurred depending on ROS production in the Stress state (PF_RP_ vs. NF_RP_; Figure 11a,b).

The results also show that the interplay between several factors can create qualitatively different loss outcomes. Changes in ROS production or scavenging can change the oxidative state of the optic nerve and trigger oxidative degeneration in large areas of the optic nerve. Regardless of the Stress state oxidative behavior, the temporal side octants suffered heavier axonal losses (Figure 11b). Nevertheless, the extent of the loss and the patterns depended heavily on the oxidative behavior in the Stress state.

If the intra-axonal H_2_O_2_ concentration was higher than outside the axon, permeability through the axonal membrane allowed the transport of H_2_O_2_ to the inter-axonal space. Conversely, if the intra-axonal H_2_O_2_ concentration was lower than outside the axon, axons were affected by extra-axonal H_2_O_2_. This oxidative connection between a neuron and its environment can be beneficial or deleterious. Under such conditions, adjacent axons create a localized oxidative environment. This oxidative interdependence can lead to axonal losses that are geometrically close, regardless of the diameters of the adjacent axons. Such topological loss patterns were called “neighborhood losses.”

Loss of neighboring axons was present regardless of the Stress state production whenever the membrane was permeable (Figure 11c). For NF_RP_ simulations, the effect was less prevalent and not as visually striking due to its prevalence in areas where small and medium diameter axons are grouped and lack of injury beyond small and medium diameter axons. (NF_RP_ simulations in Figure 11a). Areas of degeneration were many yet small (PF_RP_ marked simulations in Figure 11a). This pattern is atypical for LHON pathology [37].

In contrast, in simulations where H_2_O_2_ production increased in stressed axons (PF_RP_), there was high neighborhood loss, and in most cases, the loss was complete in each octant (PF_RP_ simulations in Figure 11). This injury pattern (Figure 11a) resembles that seen in LHON pathology [37]. Higher mito% provided more protection from injury and reduced the extent of the injury patterns in proportion to the axonal mitochondrial volume ratios (mito%). Giordano et al. [12] found that carriers of the LHON mutation who do not suffer from visual loss have increased mito% relative to LHON patients. Although there is no data correlating injury extent with mito% present in axons, our model correctly predicts that there is a mito% where the injury does not appear. In all cases, axonal degeneration started in the temporal size of the optic nerve, which also correlates with initial visual loss in LHON [16,37] (Figure 11a).

At the end of a simulation, when a steady state was reached after completion of axonal loss, the map of ROS concentration was lowest where the axonal loss was greatest, i.e., axonal loss contributed to lowering the oxidative stress in the tissue, in both the NF_RP_ and PF_RP_ paradigms (Figure 11e).

In summary, the neighborhood loss effect was stronger in the PF_RP_ paradigm, and this pattern better corresponded to LHON pathology, suggesting that no change or an increase in ROS production in stressed axons is more likely in the acute phase of LHON.

### 3.6. Glia-Produced Hydrogen Peroxide Produces Cascading Axonal Failure

Optic nerve glia are also possible sources of oxidative stress. Our previous study focused only on axonal-generated ROS [37]. Given that axonal and glial mitochondria are assumed to have Stress states, there are 4 possible combinations: axon and glial mitochondria both use NF_RP_ in the Stress state, both use PF_RP_, or they use opposite ROS production strategies. When both axonal and glial mitochondria used the same Stress state production model, the simulation results were qualitatively similar to the ones presented in the previous sections (Figure 9, Figure 10 and Figure 11). Such results are expected because adding glial elements between axons is equivalent to adding axons with a virtual membrane fully permeable to H_2_O_2_. If axonal mitochondria used PF_RP_ and glial mitochondria used NF_RP_, the results were also qualitatively close to the previous PF_RP_ studies (Figure 10 and Figure 11). Glia-produced H_2_O_2_ increases oxidative stress in all axons, exacerbating the loss experienced in PF_RP_ relative to the same simulations that do not consider glial elements.

The combination where axonal mitochondria produced less ROS in the Stress state (NF_RP_ mode) and glial mitochondria produced more ROS (PF_RP_ mode) was surprising (Figure 12 and Figure 13). Given that NF_RP_ simulations with no glia resulted in a weak neighborhood effect, we hypothesized that the addition of glia operating in PF_RP_ mode to unmyelinated axon models would increase the area of neighborhood loss, relative to the case where only axonal mitochondria were present. Indeed, we found that neighborhood effects extended over larger areas relative to the base case (i.e., NF_RP_ simulations without glia present), although the scale at which local loss occurs was significantly smaller than in the PF_RP_ cases (Figure 12a,b). If glial mitochondrial content was increased, the neighborhood effect was stronger, with local injury extending to larger groups of adjacent axons (Figure 12a). In all simulations where glial mitochondria were present, axonal loss was more prevalent in the temporal side of the model (Figure 12), as was seen when glial mitochondria were absent.

Higher glial mitochondria content will increase the overall model H_2_O_2_ concentrations. The H_2_O_2_ concentrations maps (Figure 12e) show the increase created by the glia-generated H_2_O_2_, generating a more uniform H_2_O_2_ concentration distribution across the model. Surprisingly, even under such homogeneous H_2_O_2_ concentrations, the temporal side was still more susceptible to oxidative damage, probably because of its higher density of small diameter axons. Although glia’s contribution to [H_2_O_2_] creates a more homogeneous oxidative environment, the per axon differences are still a factor (Figure 12e).

Membrane permeability had a limited effect on the effects of glial ROS production models (Figure 13), similar to cases where glial mitochondria were absent. However, the results with a 10% glia setting and lower axonal mito% proportions (4% and 5%) were independent of the membrane permeability (Figure 13). When glia mito% were decreased, significantly greater percentages of axons survived (Figure 12).

Because of membrane permeability in unmyelinated areas, H_2_O_2_ produced outside the axon will influence the axons and vice versa. Glia are present in the myelinated region but not in the areas next to the nodes [12]. Myelinated axons are impermeable to H_2_O_2_ outside the node; therefore, ROS outside the axon would not penetrate the axon. Higher H_2_O_2_ levels could affect myelination, although our models do not capture such effects. Therefore, we only simulated unmyelinated optic nerve sections in the studies of axonal-glial interactions.

Overall, H_2_O_2_ created by glia can affect axons in unmyelinated regions, but is not necessary to explain the neighborhood loss patterns exhibited by LHON patients. Glia may have a supporting role in axonal degeneration, but the main oxidative engine is likely intra-axonal mitochondria.

## 4. Discussion

The mechanism for propagation of axonal degeneration in LHON is unknown, as is the location of the initial injury within the optic nerve. We previously described a 2-dimensional coronal plane propagation algorithm that produced injury patterns which correlated with clinically observed patterns of RGC loss [37]. In the current study we expanded that work by adding 3-dimensional optic nerve section modelling, more realistic biophysical processes and parameters, use of membrane permeability, modeling of unmyelinated and myelinated axons, the addition of glia-produced ROS, and interpretation of biological vs. simulation time with respect to biology. The current work also allows the study of ROS generation and diffusion processes around the nodes of Ranvier, which are areas of more intense energy demands within the myelinated regions of the optic nerve.

We showed that H_2_O_2_ is more likely than $O_{2}^{-}$ to be the primary ROS in propagating LHON injuries, owing to its cellular membrane impermeability and its fast dismutation to H_2_O_2_. This result has therapeutic implications. Increasing peroxidase activity in the optic nerve extra-axonal space could presumably decrease the propagation of the axonal degeneration and may be more amenable to drug delivery than delivering a SOD mimetic [69,70], which would have to enter the axon.

The study modeled and explored several hypotheses regarding the influence of H_2_O_2_ in LHON injury propagation dynamics. We found that when the axonal membrane is permeable to H_2_O_2_ and oxidative stress induces larger production of H_2_O_2_, small injuries extend to neighboring axons. Our work suggests that a limited systemic insult can affect a region of the optic nerve in a process akin to a selective axotomy, triggering axonal degeneration in the respective RGCs. When some axons die, a new equilibrium point is found. This can explain why LHON rarely results in complete blindness, instead primarily producing loss of central vision and clinical findings of small axon-diameter disease.

By studying the propagation of H_2_O_2_ in models that contained unmyelinated and myelinated regions, we showed that due to their relative distance, the two regions do not influence each other oxidatively. Specifically, the origin and propagation of axonal degeneration in this simulation model is biased toward greater loss in the unmyelinated portion of the nerve rather than the myelinated nerve. In other words, it is possible that there may be early oxidative injury to axons within the optic nerve head that is of a greater degree than the retrobulbar nerve. Given that the optic nerve head is clinically visible, then if swelling or other changes could be detected before visual loss occurs, there may be a therapeutic window before the rest of the nerve is injured and axons are lost. Oxidative axon injury restricted to the unmyelinated nerve but without propagation to the rest of the nerve could cause localized loss of axonal conduction and visual loss. If conduction is restored as a result of lower ROS levels, then this could represent a mechanism by which LHON spontaneously improves in many patients.

It is known that positive feedback (i.e., the ability of an external signal to increase its production) in different systems, e.g., ROS and calcium, can result in wavelike propagation [54,71]. However, all such wave-generating models have an external wave initiation event. For example, our previous model [37] had to be triggered by an “injury stimulus.”. In contrast, the current study describes self-triggered injuries that appear because of an individual cell’s biochemistry in the context of its milieu. This may be relevant to patients with LHON who begin having visual loss soon after heavy drinking, smoking, or other stressors. LHON patients who chronically smoke are relatively protected and lose vision more insidiously [14]. It is possible that ROS levels in smokers lead to early chronic loss of small diameter axons, similar to the NF_RP_ patterns of Figure 11a. This effect might provide later protection by reducing the overall density of small axons in the optic nerve. The model would also allow simulating the effects of myelin development, mediators of systemic oxidative stress, and other parameters that could be assessed biophysically.

We found that H_2_O_2_ concentrations correlated with axonal degeneration patterns seen in LHON histological sections [37]. Higher oxidative levels were present in the temporal side of the nerve before the onset of axonal loss, which was also on the temporal side. These findings correlate well with clinical observations [2]. After axons have been affected, a new equilibrium is found where less H_2_O_2_ is produced. The degeneration of some axons and the subsequent cessation of ROS production can have a protective effect on the remaining axons. Clinical observations on children under 12 show that despite structural evidence of axonal loss, there is less functional loss and no loss later in life [72]. The proportion of axonal volume occupied by (mitochondria volume ratio; mito%) heavily influenced the extent of degeneration, in line with clinical observations [12]. Given that ROS have roles as signaling agents in the control of mitochondrial biogenesis [73], the increased mito% in LHON carriers and cases [12] could be explained by oxidative stress inducing larger mitochondrial volume ratios in the axon. It is possible that in carriers (i.e., before axonal degeneration occurs), the higher ROS levels signal locally or to the soma that more axonal mitochondria should be produced. Given that LHON mutations lead to higher ROS production [23,24], an intrinsic mitochondrial production signaling mechanism could explain the increased mito% observed in LHON cases and carriers relative to controls [12].

This study has several limitations. First, although it is well known that cellular membranes are relatively permeable to H_2_O_2_ and relatively impermeability to $O_{2}^{-}$, and the basis for the simulations that were used, the degree of permeability to H_2_O_2_ also affects the results. However, it is not known what the RGC axonal permeability to H_2_O_2_ is in either the unmyelinated or myelinated regions, and assumptions from non-axonal cell membranes may not be incorrect. Second, although the range of concentrations at which H_2_O_2_ causes oxidative stress are known [47], there is a lack of experimental data regarding the long-term effects of these concentrations on RGC axons or somas, or the time frame at which they would cause axonal degeneration. The biochemical linkage between oxidatively damaging H_2_O_2_ concentrations and mechanisms of axonal degeneration are also unknown.

Third, in silico studies are hampered by the balance between computational resources needed, simulation time, and simulation resolution. Our study is no exception. Model sizes and the model resolution were limited by the available GPU with its on-board memory and the need for simulation throughput. The methods presented in this paper can be extended to larger tissue samples. For example, extending the current work to a complete optic nerve and ultimately to an optic nerve-retina system would be worthwhile. Nevertheless, although the model can be extended spatially using more greater computational resources, there are limits to the simulation time that can be afforded. When a second in the biological life of the sample can only be processed in many seconds of computer simulation, longer-term evolving diseases such as LHON cannot be simulated at the time scales of the illness. On the other hand, we do not know how long the pathophysiological process takes, and are limited by patient symptoms, measurements of visual loss, clinical observation of disc edema and other retinal findings, and detection of axon and ganglion cell/inner plexiform layer loss in the retina.

Fourth, the timelines of the simulated processes are more rapid than the timeline of visual loss in LHON. Given that the simulation results are based on realistic biological parameters, the difference in timing suggests a possible intermediate mechanism triggered by ROS, which subsequently results in axonal dysfunction and degeneration, and eventually results in somal apoptosis. Such a mechanism could be a delay until the axon becomes dysfunctional, based on a biochemical, inflammatory, or other mechanisms.

Finsterer et al. [74] reviewed journal-based reports of diseases seen in LHON patients which affect other organs. These data could support the extension of our hypothesis of ROS-induced damage in LHON patients to other organs, providing a comprehensive explanation of the observed organ problems. For example, a study by Orssaud [75] reports abnormal cardiac function in LHON patients and recommends cardiac evaluation for such sufferers. The linkage of ROS to the most common cardiac problems, such as arrhythmia [76], sudden cardiac death [77], and the homoplastic nature of the LHON genetic defects, provides support for oxidative stress as a possible explanation for the misfunction of various organs in LHON patients.

No similar data regarding carriers were found in the literature. An interesting data point would be to see if non-visual diseases occur more or less frequently in carriers vs. controls or vs. LHON cases.

In summary, a more comprehensive model of ROS in LHON and the effects on the propagation of axonal degeneration has been developed. The model is more realistic anatomically, biophysically, and biochemically, and has implications for the pathophysiology of the disease and the development of therapeutic approaches. Future studies could address biochemical feedback loops between RGC and glia, the role of other ROS, the specific anatomy of the optic nerve head, and use these refinements to further advance the detection and treatment of LHON in those at risk.

## Figures and Tables

**Figure 1 biomolecules-12-01411-f001:**
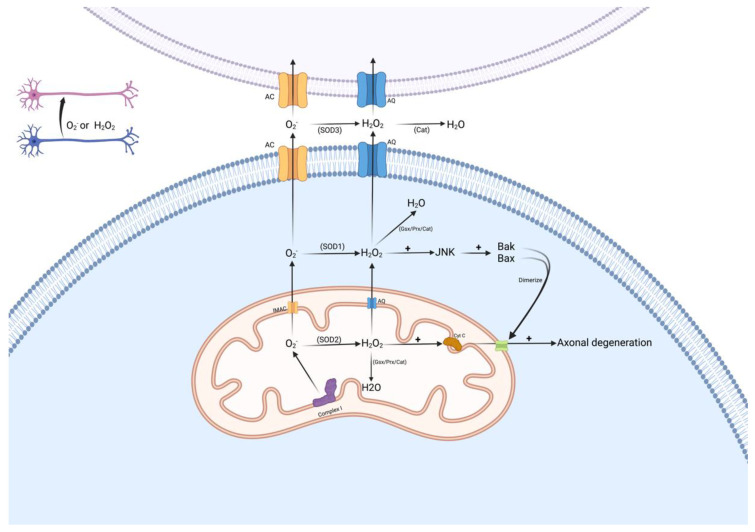
The biochemical model uses two ROS species, $O_{2}^{-}$ and H_2_O_2._ Mitochondria are considered the main source of $O_{2}^{-}$. The dismutation of $O_{2}^{-}$ results in H_2_O_2_. While there are other H_2_O_2_ sources, the current study explores only the effects of H_2_O_2_ produced from $O_{2}^{-}$. Facilitated diffusion of H_2_O_2_ is conducted through aquaporin (AQ) channels. Low quantities of $O_{2}^{-}$ diffuse through anionic channels (AC), such as subsets of chloride-conducting channels.

**Figure 2 biomolecules-12-01411-f002:**
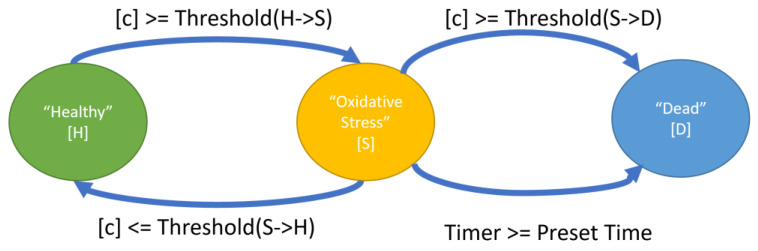
State-dependent ROS production and scavenging rates were assigned to individual voxels: Healthy (H), Oxidative Stress (S) or Dead (D). State changes were dynamic, based on either threshold conditions or triggered by the expiration of timers.

**Figure 3 biomolecules-12-01411-f003:**
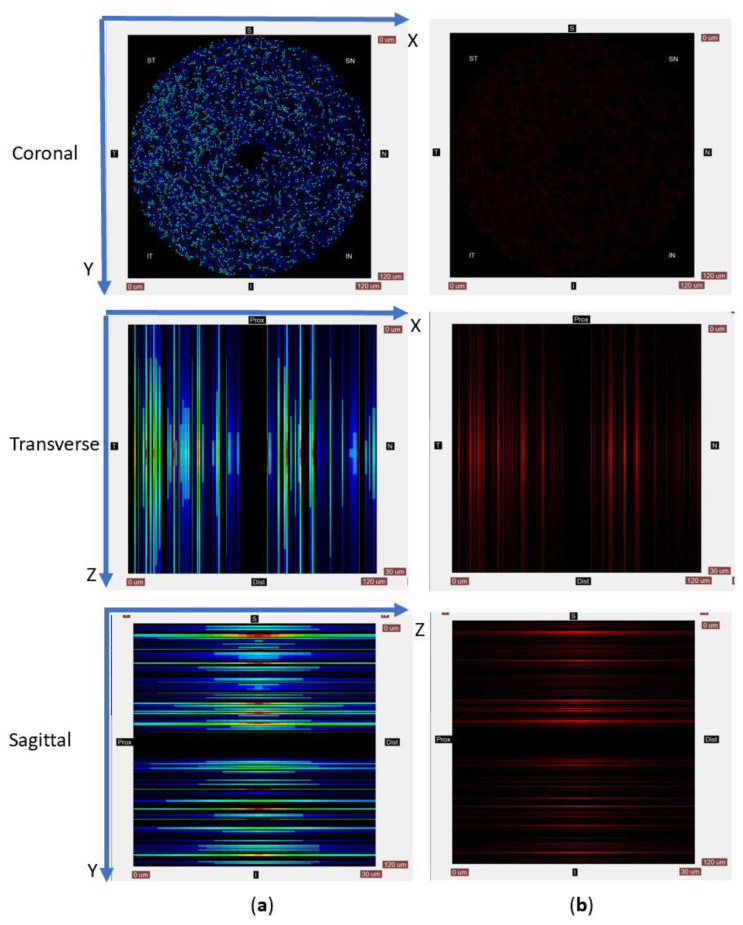
ROS concentrations can be visualized using a heatmap (**a**) or single-color intensity (**b**). Sample physical dimensions are mapped to the same number of pixels, resulting in square images even if the sample cut would be a rectangle. In all pictures, black denotes a zero concentration of species. The top of the transverse images represents the model’s proximal side, while the bottom of the image shows the model’s distal side. The transverse view’s left side shows the temporal (T) octant, while the right side of the image corresponds to the nasal (N) octant. In the sagittal views, the picture’s left shows the proximal plane, while the picture’s right shows the distal part. The image top corresponds to the superior (S) octant and the bottom of the image to the inferior (I) octant. Superimposed on the sample is a Cartesian system of coordinates. The coordinate system origin is in the left corner of the coronal plane. The X-axis runs left to right along the top line of the coronal cut, while the Y-axis runs from top to bottom in the same view. Thus, the coronal plane cut displays the XY plane. The Z-axis runs along the proximal to distal direction, with the origin on the axis at the proximal side.

**Figure 4 biomolecules-12-01411-f004:**
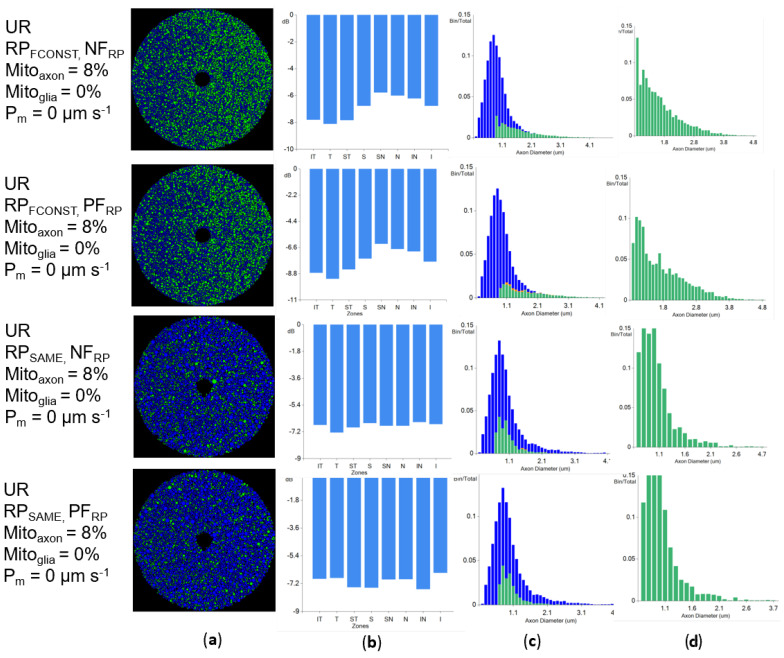
Lack of membrane permeability to ROS species creates evenly distributed zonal loss in the unmyelinated region (UR). NF_RP_ and PF_RP_ simulations have similar end states when the membrane is impermeable: (**a**) Coronal view of axonal state with healthy axons in green and dead axons in blue; (**b**) Coronal octant axonal loss is expressed in dB; (**c**) The distribution of axonal diameters, with green bars representing healthy axons and the blue bars representing the dead axons; (**d**) Diameter distributions for surviving axons.

**Figure 5 biomolecules-12-01411-f005:**
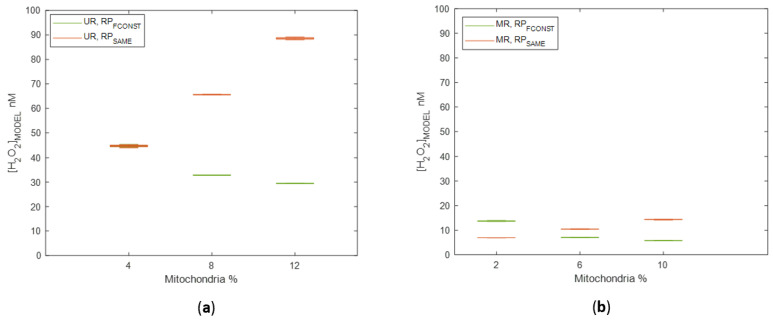
Sample H_2_O_2_ concentrations ([H_2_O_2_]) depend on mitochondrial volume ratio for ROS production models RP_SAME_ and RP_FCONST_: (**a**) Mean H_2_O_2_ concentrations for simulations of unmyelinated regions (UR); (**b**) Mean H_2_O_2_ concentrations for simulations of myelinated regions (MR).

**Figure 6 biomolecules-12-01411-f006:**
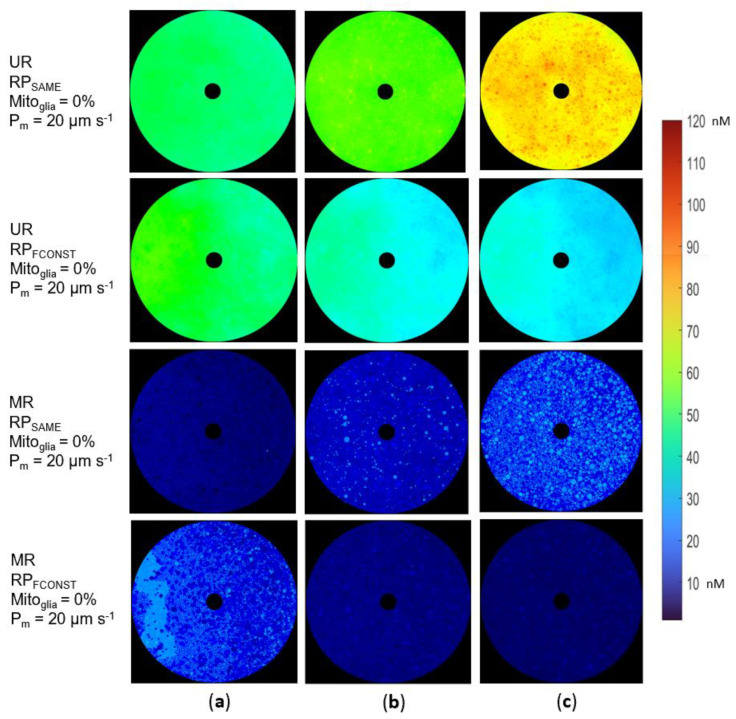
Coronal plane views of simulations show [H_2_O_2_] gradients from the RP_SAME_ and RP_FCONST_ ROS production models in both myelinated (MR) and unmyelinated regions (UR): (**a**) [H_2_O_2_] variations at an axonal mitochondria volume ratio (mito%) of 4%; (**b**) [H_2_O_2_] values and gradients when the mito% doubles (8%); (**c**) [H_2_O_2_] values and gradients when the mito% is 12%. [H_2_O_2_] was highest in simulations of unmyelinated regions which used the RP_SAME_ production hypothesis, with many localized values close to the oxidative stress threshold.

**Figure 7 biomolecules-12-01411-f007:**
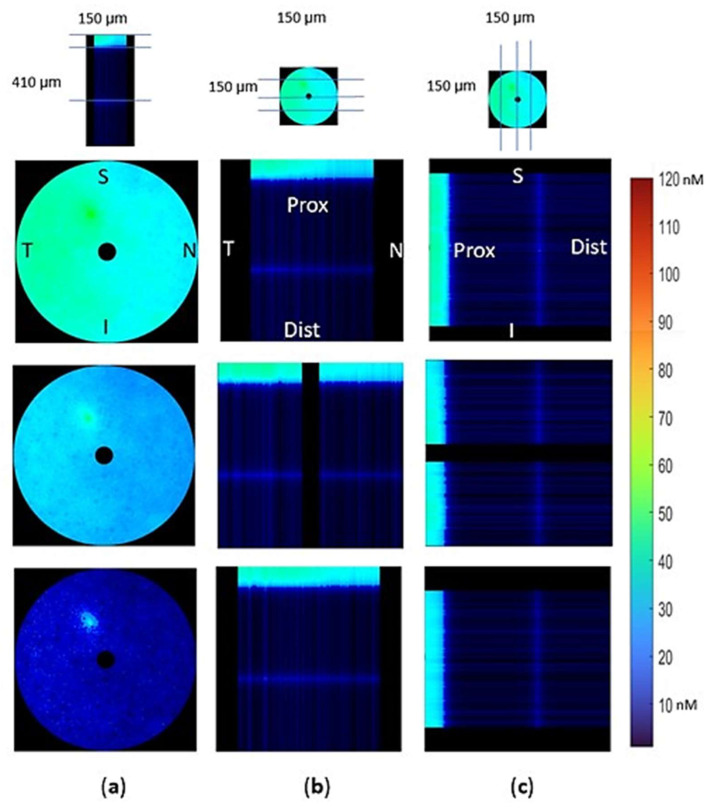
Models with unmyelinated and myelinated regions show distinct [H_2_O_2_] patterns in the two regions, with higher concentrations of H_2_O_2_ in the unmyelinated region (P_m_ = 20 µm s^−1^, uniform scavenging along the Z axis). The intensity map values are in nM. (**a**) Coronal plane [H_2_O_2_] map at 3 places along the Z axis; (**b**) Transverse views show large [H_2_O_2_] differences along the Z axis. (**c**) Sagittal views of [H_2_O_2_] demonstrate that peak concentrations are in the unmyelinated region, similar to what is seen in the transverse views.

**Figure 8 biomolecules-12-01411-f008:**
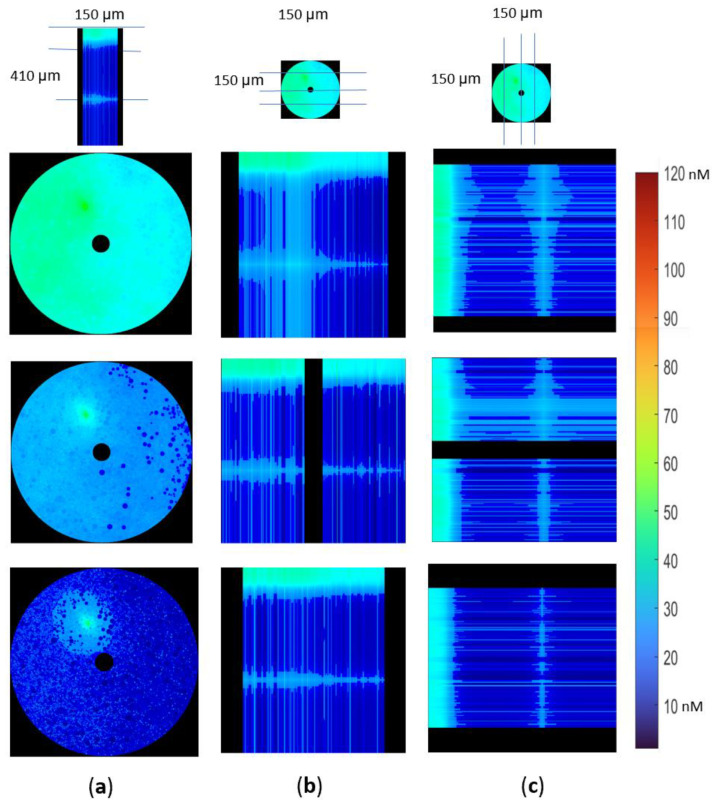
Models with unmyelinated and myelinated regions have peak [H_2_O_2_] in the unmyelinated regions and higher concentrations close to the nodes of Ranvier than in the internodal regions when scavenging in the internodal regions is lower than in the rest of the model (P_m_ = 20 µm s^−1^, 30% reduction in scavenging capacity in the internodal regions). (**a**) The coronal plane cuts show the [H_2_O_2_] at 3 distinct Z locations along the axon; (**b**) Transverse views of [H_2_O_2_] gradients in the unmyelinated and myelinated regions; (**c**) Sagittal view of [H_2_O_2_] and the differences between unmyelinated and myelinated regions.

**Figure 9 biomolecules-12-01411-f009:**
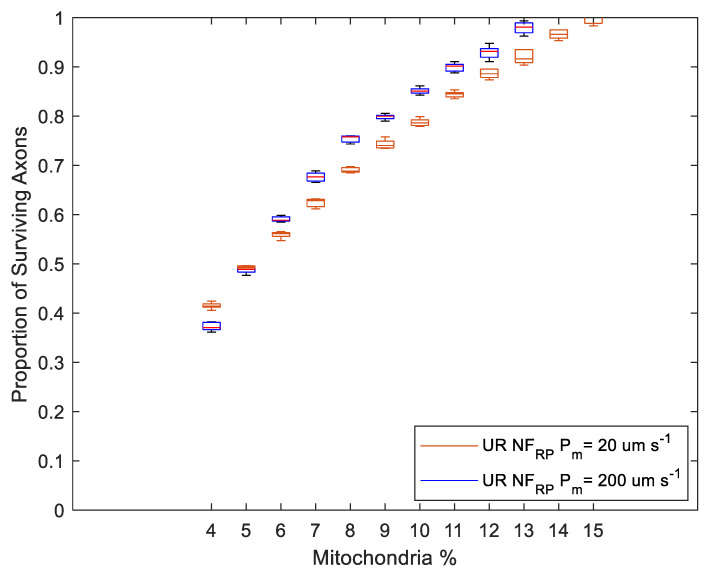
Axonal survival increases proportionally with the mitochondrial volume ratio (mito%) when the H_2_O_2_ production rate in the Stress state is lower than in the Healthy state (NF_RP_). Mito% varied between 4 and 20%. Myelinated region models experienced no loss. Simulations using unmyelinated and myelinated models were performed with the same biochemical parameters.

**Figure 10 biomolecules-12-01411-f010:**
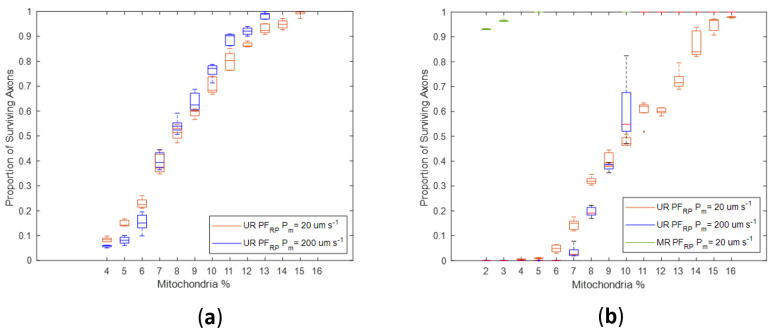
The proportion of surviving axons when the H_2_O_2_ production rate in the Healthy state is lower or equal than in the Stress state (PF_RP_) increases as the mitochondrial volume ratio (mito%) increases. Mito% varied between 4 and 20% for unmyelinated region simulations and 2 to 6% for myelinated region simulations. (**a**) The proportion of surviving axons as a function of mito%, when production rates in the Healthy and Stress states are equal; (**b**) The proportion of surviving axons as a function of mito% when the H_2_O_2_ production rates in the Stress state are 30% higher than in the Healthy state.

**Figure 11 biomolecules-12-01411-f011:**
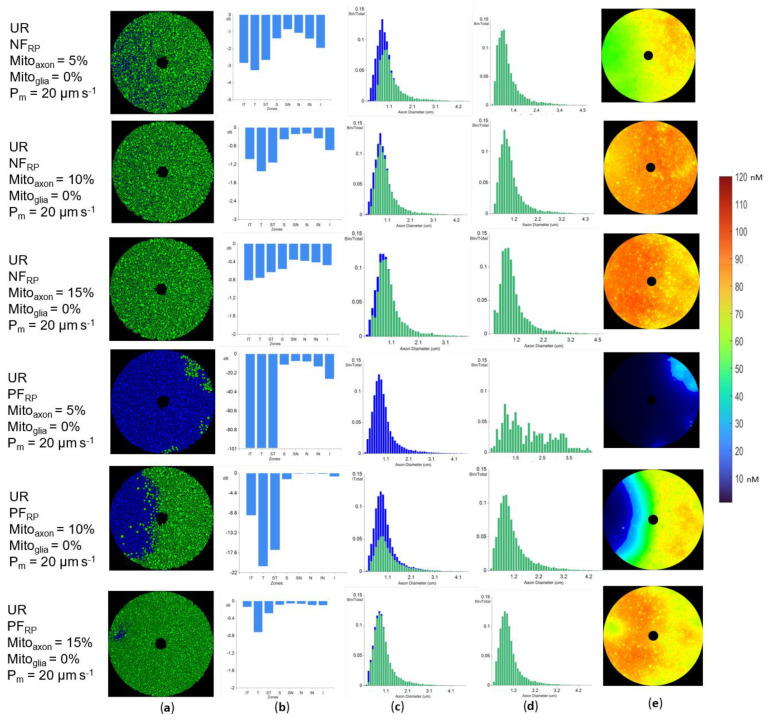
Higher mitochondrial volume ratios (mito%) reduce the injury extent in NF_RP_ and PF_RP_ simulations: (**a**) Coronal view of the axon states at the end of simulations; (**b**) Coronal view octant loss, expressed in dB; (**c**) Axonal diameter distributions, with stacked bars for each diameter. Green bars represent Healthy axons, while blue bars represent Dead axons; (**d**) Axonal diameter distribution for surviving axons only; (**e**) Coronal view of [H_2_O_2_] at the end of the simulations. Bar values are in nM.

**Figure 12 biomolecules-12-01411-f012:**
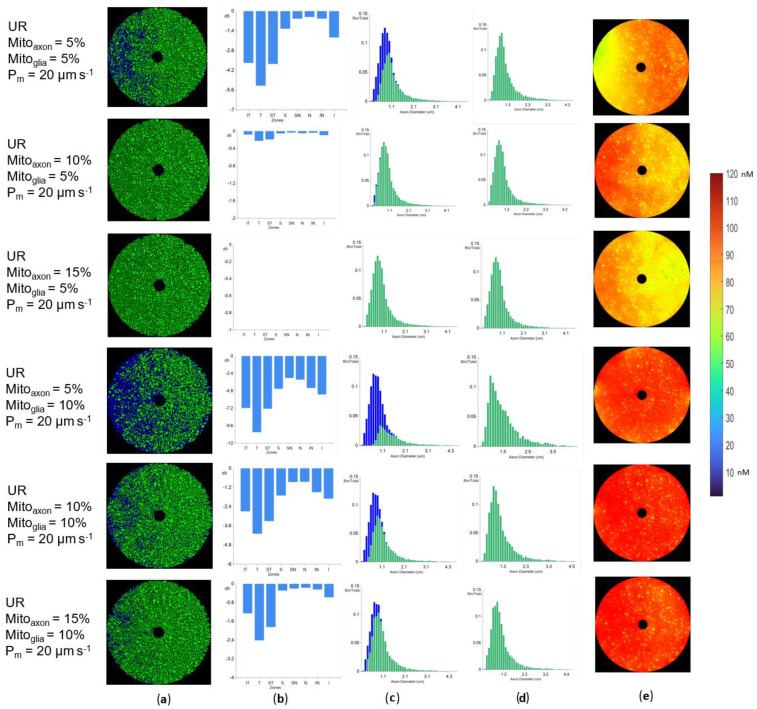
Injury extent and axonal loss distributions in glia-enabled simulations depend on the axonal mitochondrial volume ratios (mito%) and glial mitochondrial volumes: (**a**) Coronal plane view of axonal states at the end of the simulations; (**b**) Coronal plane octant loss, in dB; (**c**) Axonal diameter distributions with details of their state, blue bars for Dead axons and green bars for Healthy ones; (**d**) Histogram of surviving axons diameters; (**e**) Coronal view of the [H_2_O_2_] at the end of each experiment. Heatmap bar values are in nM.

**Figure 13 biomolecules-12-01411-f013:**
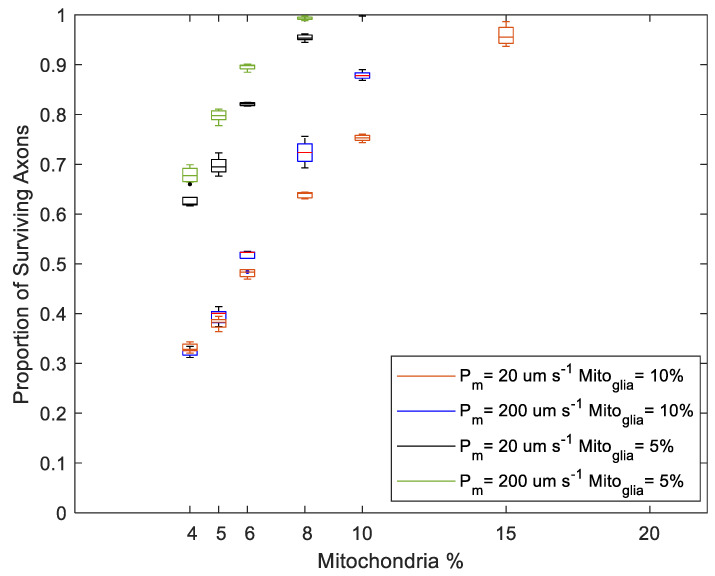
Proportion of axonal survival when axons and glia interact via H_2_O_2_ increased as the mito% increased. Axonal mitochondria operated in NF_RP_ mode while glia mitochondria were in PF_RP_ mode. Two membrane permeability settings were used (P_m_ = 20 µm s^−1^ and P_m_ = 200 µm s^−1^), as well as two settings for the glial mitochondrial content (5% and 10%) for a total of 4 data sets. For each pair of settings P_m_ and Mito_glia_, mito% was varied between 4 and 20%.

## Data Availability

The data presented in this study are available in https://github.com/dwlambiri/ros-simulator-paper/LHON-Form/Figures (accessed on 29 August 2022).

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
