# Peer review of "Modeling Reactive Oxygen Species-Induced Axonal Loss in Leber Hereditary Optic Neuropathy"

_biomolecules, 2022, doi:10.3390/biom12101411_

Round 1

Reviewer 1 Report

The authors present a cogent and fascinating description of a hypothesis tested with an "in-silico" platform for modeling.  I appreciate the novelty of the modeling and the excellence of the analysis of their simulations.  But there should be greater clarity that the risk index equations that lie at the heart of the axon size contribution to the risk index is not novel.  

1)  To begin with, on page 2, line 70, it would be helpful to add another reference to this story that includes the concept of risk index as inversely proportional to axon diameter.  Alfredo A Sadun, Chiara La Morgia, Valerio Carelli.  Mitochondrial optic neuropathies: our travels from bench to bedside and back again. Clin Exp Ophthalmol. 2013 Sep-Oct;41(7):702-12. doi: 10.1111/ceo.12086. Epub 2013 Apr 11.

 Further, that the small axons are the first to die in LHON was more completely described in Sadun AA, Win PH, Ross-Cisneros FN, Walker SO, Carelli V.  Leber's hereditary optic neuropathy differentially affects smaller axons in the optic nerve.  Trans Am Ophthalmol Soc. 2000;98:223-32; discussion 232-5. 

2)  It's important to say that these and other works, including the Pan et al article that they reference, are all from the Sadun lab.  Pan was a medical student.  It would be more appropriate to say something like this at line 76:  " which led the Sadun group (refs 16 and the two new references in point 1) to hypothesize that ..."

3) After line 82, it would be useful to add a new and very relevant concept.  Not only do fibers die as a wave, small axons to large (by zones), but they do so almost all at once.  The kinetics are very interesting.  Any explanation (very much including their own with O2 as the vector) need to link axon degeneration to nearest neighbor, to zones, and to speed.  

4)  About line 85-92:  There is a very relevant and important reference and discussion missing here.  This ref. in PNAS both demonstrates that ATP production is NOT lowered in a faithful mouse model of ND6 dysfunction, but that the ROS levels are much high in synaptosome analysis.  Chun Shi Lin, Mark S Sharpley, Weiwei Fan, Katrina G Waymire, ….., Douglas C Wallace.  Proc Natl Acad Sci U S A. 2012 Dec 4;109(49):20065-70. doi: 10.1073/pnas.1217113109. Epub 2012 Nov 5.  Mouse mtDNA mutant model of Leber hereditary optic neuropathy.

5)  Line 105.  ROS production increases, but by how much?  Wouldn't quantitation really add a lot to the argument?  

6)  Line 110.  Therapeutic measures to address ROS through antioxidants have been addressed, for example by Pisano A, Preziuso C, Iommarini L, Perli E, Grazioli P, …. Giordano C. Targeting estrogen receptor β as preventive therapeutic strategy for Leber's hereditary optic neuropathy. Hum Mol Genet. 2015 Dec 15;24(24):6921-31. doi: 10.1093/hmg/ddv396. Epub 2015 Sep 26. PMID: 26410888

7) line 259.  Why talk about the optic nerve size?  What matters are the sizes of the axons.   The range of human optic nerve axons by EM measures are 0.4 - 2.2 um.  Higher numbers are sometimes reported with automated counts that undercount smaller axons that are missed.  The average is closer to 1.0 than 1.5 um.   

8)  Mitochondrial dimensions are also systematically overestimated.  Most mitochondria are about 0.5 um though, obviously smaller in the very smallest axons where the mitochondrial diameters are about 0.3 um.  Their "length" is not relevant and varies by the fission/fusion ratios often dependent on oxidative stress.  

9)  The equations (9 and 10 and 11) between lines 381 and 390 are equivalent to the stress index ( NFL-SI = ASA/AV = L2πR/LπR2) that was previously published and described in Fig, 2 in their reference 16.  At minimum, they should point this out and reference the equivalence.  

10)  Lines 845-850.  About here, there are two additional points that could be made that would buttress their analysis.  The first is that in children under 12, the amount of structural damage from LHON is about the same, but the functional damage is less.  But the children do well as there is no further loss.  This suggests, as the present work would suggest, that clearing out many of the smaller axons is protective.  Here is one of several references: Barboni P, Savini G, Valentino ML, La Morgia C, …. Carelli V. Leber's hereditary optic neuropathy with childhood onset. Invest Ophthalmol Vis Sci. 2006 Dec;47(12):5303-9. doi: 10.1167/iovs.06-0520.

 11)  An even stronger point is that in an atypical version of LHON, the carriers are older, lose vision more insidiously, and are paradoxically protected by long term cigarette smoking.  It is likely that this high ROS exposure has culled out some of the smallest fibers allowing for longer term protection from the subacute and dramatic loss of vision.  It's equivalent to burning the brush to avoid forest fires.  Here's the reference: Carelli V, d'Adamo P, Valentino ML, La Morgia C…., Sadun AA. Parsing the differences in affected with LHON: genetic versus environmental triggers of disease conversion.  Brain. 2016 Mar;139(Pt 3):e17. doi: 10.1093/brain/awv339. Epub 2015 Dec 10.  BTW, others, including Kirkman et al have observed the phenomenon but offered no explanation.  

The conclusions are cogent and supported by their model data and analysis.  This is a valuable and original contribution to the literature.  

Reviewer 2 Report

This is an interesting manuscript in which computational biology and three-dimensional modeling of myelinated and unmyelinated optic nerve segments are used to evaluate various underlying assumptions regarding the nature of reactive oxygen species formation, scavenging and propagation as an explanation for the pattern of axon loss observed in LHON. The authors' work should certainly attract the interest of scientists and neuro-ophthalmologists who have puzzled over the unique clinical characteristics of LHON, such as its variable penetrance and its propensity to affect retinal ganglion cells of the nasal macula. I have several comments that I would like the authors to address:

1. I was somewhat surprised by the focus on H2O2 given one of the author’s 2019 Scientific Reports article that models superoxide propagation in the context of LHON. Superoxide as a major player is dismissed early on in the present manuscript because of its membrane impermeability as a charged molecule, but this was not considered problematic in the author's earlier modeling. The diagram shown in Figure 1, depicts both superoxide and H2O2 leaving and entering cells through some form of channel (denoted “AC” and “AQ” respectively, but without defining what these abbreviations are). If such channels exist, then it seems that they may be just as relevant as the membrane permeability of individual reactive oxygen species and might therefore need to be included in the model as well. At the very least, it would seem harder to eliminate superoxide from consideration if a means of passage from one cell to another indeed exists.

2. While I would not expect it to be modeled here, is it not possible that the site of pathology could be in the axons of the retinal nerve fiber layer (RNFL), before they converge at the optic nerve head? The RNFL represents a relatively lengthy region in which the axons are unmyelinated and therefore conduct the action potential in a very energetically expensive way. Perhaps damage from reactive oxygen species could propagate and spread between axons in the RNFL rather than in the optic nerve; if anatomic features of the papillomacular bundle make it particularly susceptible, this would explain why the temporal optic nerve suffers the most damage early in LHON.

3. Can the authors comment on whether their simulation takes into account the temporal progression of retinal ganglion cell loss in LHON. Vision loss may develop over a period of weeks or even months in LHON, and RNFL thinning may not level off for up to a year. Does their model allow for such a slow propagation of a stressor between axons? Or, if the model hinges on achieving a steady state of reactive oxygen species, does the simulation necessarily culminate within a short period of tme (e.g. minutes)? I would agree that reactive oxygen species likely play a key role in RGC degeneration in LHON, but could it be that the propagation of axon death is due to the death of one axon causing neuroinflammation (or releasing some type of toxic mediatory distinct from ROS) that then induces stress in neighboring axons, causing them to be more susceptible to their own ROS?

4. In Results section 3.1, would you say that the selection of 10-2 nM of superoxide as the toxicity threshold is entirely arbitrary? It seems that you question whether it is even physiologically achievable.

5. Rather than “high diameter” or “low diameter”, the adjectives “larger” or “smaller” may be more appropriate for describing diameters.

6. Is the paragraph starting at line 446 reporting the same data as two paragraphs before (starting on line 428?

7. Figures 4 through 6: somewhere it should be made clear that UR is “unmyelinated region” and MR is “myelinated region.”

8. Line 554: can you explain your reasoning for assuming that ROS scavenging capacity might be 30% lower in myelinated portions of the axon? I assume it was selected arbitrarily, but in that event, why not 30% higher? If scavenging is dependent on cytoplasmic volume, then with less volume being taken up by mitochondria in myelinated portions of the axon, then perhaps ROS scavenging would be increased?

9. End of section 3.5 (paragraph beginning on line 734): you mention that at the steady-state, after completion of axonal loss, the ROS concentration is lowest where axonal loss was greatest. This is described as “paradoxical”, but it seems to be quite predictable, since if there are few axons, where will the ROS come from? This is another place where I wonder whether your model gives any information on when this “steady-state” is achieved? If the steady-state requires that certain axons have already died off, doesn’t it matter when the axons die off? Rapid die-off might impose less ROS on surrounding axons than a slower die-off.

10. Discussion: paragraph starting on line 841: In the last two sentences of the paragraph, it states: “In other words, it is possible that there may be early loss of axons within the optic nerve head that is of a greater degree than the retrobulbar nerve, and this may increase the time for therapeutic interventions. It may also be a potential mechanism by which LHON spontaneously improves in many patients.” Both of these sentences sound quite intriguing, but they need much more explanation.  I cannot understand the reasoning for either, as far as increasing the therapeutic window or explaining spontaneous recovery go.
